# Efficient Cell Painting Image Representation Learning via Cross-Well Aligned Masked Siamese Network

**Pin-Jui Huang**[*]
Smart Group Solution Corp.
jefferyhuang@sgsc.ai

**Yu-Hsuan Liao**[*]
Smart Group Solution Corp.
samliao@sgsc.ai

**SooHeon Kim**
OmixAI Co. Ltd.
shk@omixai.com

**NoSeong Park**
KAIST
noseong@kaist.ac.kr

**JongBae Park**
Kyunghee Univ.
OmixAI Co. Ltd.
jbp@khu.ac.kr

**DongMyung Shin**
OmixAI Co. Ltd.
Oncocross Co. Ltd.
shinsae11@gmail.com

## Abstract

Computational models that predict cellular phenotypic responses to chemical and genetic perturbations can accelerate drug discovery by prioritizing therapeutic hypotheses and reducing costly wet-lab iteration. However, extracting biologically meaningful and batch-robust cell painting representations remains challenging. Conventional self-supervised and contrastive learning approaches often require a large-scale model and/or a huge amount of carefully curated data, still struggling with batch effects. We present Cross-Well Aligned Masked Siamese Network (CWA-MSN), a novel representation learning framework that aligns embeddings of cells subjected to the same perturbation across different wells, enforcing semantic consistency despite batch effects. Integrated into a masked siamese architecture, this alignment yields features that capture fine-grained morphology while remaining data- and parameter-efficient. For instance, in a gene-gene relationship retrieval benchmark, CWA-MSN outperforms the state-of-the-art publicly available self-supervised (OpenPhenom) and contrastive learning (CellCLIP) methods, improving the benchmark scores by +29% and +9%, respectively, while training on substantially fewer data (e.g., 0.2M images for CWA-MSN vs. 2.2M images for OpenPhenom) or smaller model size (e.g., 22M parameters for CWA-MSN vs. 1.48B parameters for CellCLIP). Extensive experiments demonstrate that CWA-MSN is a simple and effective way to learn cell image representation, enabling efficient phenotype modeling even under limited data and parameter budgets. The source code for CWA-MSN is available at code link.

## 1 Introduction

Computational modeling of cellular responses to perturbations is a promising strategy for drug discovery (Noutahi et al., 2025; Liu et al., 2025; Navidi et al., 2025), predicting therapeutic effects and mechanisms of action (Tanaka et al., 2025), reducing costly wet lab experiments (Bunne et al., 2024; Adduri et al., 2025), and accelerating screening to validation (Stokes et al., 2020). High-content screening (HCS) (Bickle, 2010) enables automated acquisition of cell painting images (Starkuviene & Pepperkok, 2007), generating rich datasets for phenotype-driven modeling (Nierode et al., 2016).

Extracting meaningful representations from these images is challenging. CellProfiler (Stirling et al., 2021) uses handcrafted features to enable discoveries (Boutros et al., 2015; Ariffin, 2023) but is sensitive to batch effects (Arevalo et al., 2024b) and cannot capture complex phenotypic variations (Kim et al., 2025). Self-supervised learning (SSL) (He et al., 2020; Chen et al., 2020; Chen & He, 2021; Caron et al., 2021; He et al., 2022) can learn morphological features (Kraus et al., 2024;

---

[*]Equal contribution

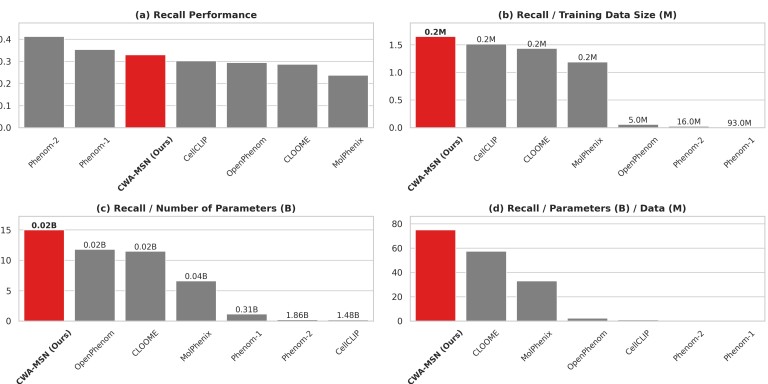

Figure 1: **Comparison of methods based on gene-gene interaction benchmark over multiple efficiency metrics:** (a) benchmark results measured as recall, (b) recall normalized by training data size (per million images), (c) recall normalized by number of parameters (per billion), and (d) recall normalized by the product of both training data size and number of parameters. Our method for each metric plot is highlighted in red. Annotated values indicate either training dataset size (M) or number of parameters (B). Except (a), CWA-MSN is top-performing, showcasing its data- and parameter-efficient learning for cell representation.

Kenyon-Dean et al., 2024) but requires large models (Dosovitskiy et al., 2020) and curated datasets. Weakly supervised or contrastive methods use proxy labels for data-efficient training (Moshkov et al., 2024; Caicedo et al., 2018; Lu et al., 2025; Sanchez-Fernandez et al., 2023; Fradkin et al., 2024; Bushiri Pwesombo et al., 2025), yet remain sensitive to batch effects.

We introduce the Cross-Well Aligned Masked Siamese Network (CWA-MSN), a novel representation learning framework for cell painting images. Unlike conventional self-supervised methods, CWA-MSN leverages weak perturbation labels to align representations across wells of the same perturbation. This cross-well alignment enforces robust semantic consistency, preserving biologically meaningful relationships instead of confounding batch effects introduced by experiment technical factors. By integrating this alignment into a masked siamese network (Assran et al., 2022), CWA-MSN substantially improves the capture of phenotypic relationships while maintaining high data and parameter efficiency.

Extensive experiments demonstrate that CWA-MSN consistently outperforms existing approaches in biological relationship retrieval tasks, particularly gene–gene and compound–gene associations (Kraus et al., 2025). On gene–gene interaction benchmarks, CWA-MSN surpasses the state-of-the-art self-supervised method OpenPhenom (Kraus et al., 2024) and the weakly supervised Cell-CLIP (Lu et al., 2025) by 29% and 9%, respectively. These gains are achieved with significantly reduced resources—0.2M versus 2.2M training images for OpenPhenom, or 22M versus 1.48B model parameters for CellCLIP. Fig. 1 illustrates CWA-MSN's advantages in terms of model size, training data, and performance.

## 2 RELATED WORK

### 2.1 SELF-SUPERVISED LEARNING FOR CELL PAINTING IMAGES

Self-supervised learning (He et al., 2020; 2022; Chen et al., 2020; Chen & He, 2021; Caron et al., 2021) has shown promise for microscopy images, but transferring methods from natural images to HCS data can be challenging. For example, DINO (Caron et al., 2021) relies on augmentations designed for natural images, limiting effectiveness on HCS (Doron et al., 2023; Kim et al., 2025; Kraus et al., 2024). Masked image modeling like MAE (He et al., 2022) reduces augmentation dependency and has successfully retrieved biological relationships (Kraus et al., 2024; Kenyon-Dean et al., 2024). However, these methods require massive compute (256 H100 GPUs (Kenyon-Dean et al., 2024)) and large datasets (93M images (Kenyon-Dean et al., 2024)). We propose a more data- and parameter-efficient approach achieving competitive performance.

## 2.2 WEAKLY SUPERVISED AND CONTRASTIVE LEARNING FOR CELL PAINTING IMAGES

Weakly supervised and contrastive learning (Yu et al., 2025; Bao et al., 2023) leverage proxy labels to train image encoders (Moshkov et al., 2024; Caicedo et al., 2018; Sanchez-Fernandez et al., 2023; Lu et al., 2025; Fradkin et al., 2024; Bushiri Pwesombo et al., 2025). SemiSupCon (Bushiri Pwesombo et al., 2025) aligns replicative features via contrastive learning but ignores cross-well, plate, and batch effects, while CellCLIP (Lu et al., 2025) uses text-based signals. SSLProfiler (Dai et al., 2025) aligns site-level images with DINOv2 but assumes identical perturbations per well, missing cross-plate variation (Moshkov et al., 2024). Our method explicitly aligns wells with the same perturbation across plates with prototype-based objective, reducing confounding from batch effects without proxy labels.

## 3 CROSS-WELL ALIGNED MASKED SIAMESE NETWORK

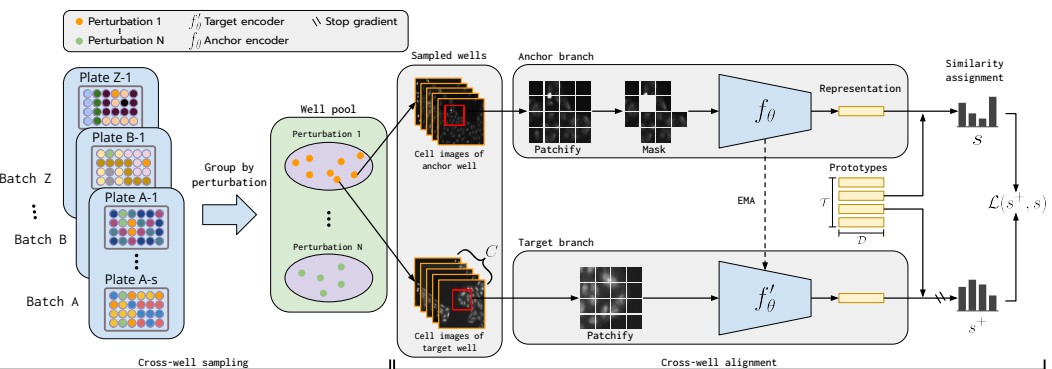

Figure 2: **Overview of CWA-MSN framework:** The framework is composed of two parts, cross-well sampling and cross-well alignment. Cross-well sampling selects cell images under the same perturbation from different wells across batches and plates to serve as an implicit data augmentation strategy. Cross-well alignment utilizes a masked siamese network to align anchor and target well representations by matching their prototype-based similarity distributions.

### 3.1 PROBLEM STATEMENT

HCS experiments produce hierarchically structured cellular imaging data. A batch corresponds to a set of plates (e.g., Batch A in Fig. 2) processed under uniform experimental conditions, and each plate contains multiple wells (e.g., 96 or 384) with replicates of cells under a specific perturbation (e.g., six wells per perturbation).

Batch effects from instrumentation, imaging, sample preparation, and technical noise can obscure true perturbation signals. We address this with a data-efficient approach using cross-well alignment and masked Siamese network (MSN) learning, which we select over alternatives such as DINOv2 because its prototype-aligned objective provides a stable signal with minimal reliance, avoiding distortion of subtle perturbation-dependent features in microscopy.

### 3.2 CROSS-WELL SAMPLING

In CWA-MSN, cross-well images of the same perturbation across plates and batches are used as implicit data augmentation. Let $P = \{p_1, \ldots, p_N\}$ denote $N$ perturbations, each associated with a set of wells:

$$W_i = \{w_1^{(i)}, \ldots, w_{M_i}^{(i)}\}, \quad w \in \mathbb{R}^{C \times H \times W},$$

where $M_i$ is the number of wells for perturbation $p_i$, $C$ is the number of channels, and $H \times W$ is the spatial dimension.

For sampling, a perturbation $p \in P$ is randomly selected, and two distinct wells under $p$ are chosen:

$$w_a^p, w_t^p \in W_p, \quad w_a^p \neq w_t^p,$$

where $w_a^p$ and $w_t^p$ are the anchor and target wells, potentially from the same or different plates/batches (Fig. 2).

### 3.3 CROSS-WELL ALIGNMENT VIA MASKED SIAMESE NETWORK

CWA-MSN combines cross-well sampling with a masked siamese network (Assran et al., 2022). Unlike MAE, which reconstructs masked regions of a single image, MSN aligns representations of two images (here, cross-well pairs) using masked and unmasked views via prototype-based learning.

For the anchor well $w_a^p$, $V_a$ augmented views are generated using random crops and flips:

$$\mathbf{X}_a^{(p)} \in \mathbb{R}^{V_a \times C \times H \times W}.$$

The target well $w_t^p$ is represented by a single augmented view:

$$\mathbf{X}_t^{(p)} \in \mathbb{R}^{1 \times C \times H \times W}.$$

A mini-batch is formed by stacking anchor and target views for perturbations $P_B \subset P$ with $|P_B| = B$:

$$\mathbf{X}_a = \{\mathbf{X}_a^{(p)}\}_{p \in P_B} \in \mathbb{R}^{B \times V_a \times C \times H \times W}, \quad \mathbf{X}_t = \{\mathbf{X}_t^{(p)}\}_{p \in P_B} \in \mathbb{R}^{B \times 1 \times C \times H \times W}.$$

The anchor view $\mathbf{X}_a$ is patchified and masked with ratio $\alpha$, while $\mathbf{X}_t$ is only patchified. Embeddings are computed via anchor and target encoders:

$$z = f_\theta(\mathbf{X}_a) \in \mathbb{R}^{B \times V_a \times \mathcal{D}}, \quad z^+ = f_\theta'(\mathbf{X}_t) \in \mathbb{R}^{B \times 1 \times \mathcal{D}}, \tag{1}$$

with representation dimension $\mathcal{D}$.

With a set of prototype embeddings $\mathrm{O} \in \mathbb{R}^{\mathcal{T} \times \mathcal{D}}$, where $\mathcal{T}$ is the number of prototypes, we compute the similarity assignment scores as:

$$s = \mathrm{sim}(\mathrm{O}, z) \in \mathbb{R}^{B \times V_a \times \mathcal{T}}, \quad s^+ = \mathrm{sim}(\mathrm{O}, z^+) \in \mathbb{R}^{B \times 1 \times \mathcal{T}}. \tag{2}$$

where $\mathrm{sim}(\cdot, \cdot)$ denotes the normalized cosine similarity.

The model is trained to align anchor and target similarity distributions with an auxiliary mean-entropy term to prevent collapse:

$$\mathcal{L}(s^+, s) = \lambda_1 \, CE(s^+, s) + \lambda_2 \frac{1}{B\mathcal{T}} \sum_{j=1}^{B} \sum_{m=1}^{\mathcal{T}} s_{j,m}, \tag{3}$$

where $CE(\cdot, \cdot)$ is cross-entropy and $\lambda_1, \lambda_2$ balance the terms. The target encoder $f_\theta'$ is updated via exponential moving average (EMA) of $f_\theta$ (Fig. 2).

### 3.4 IMPLEMENTATION DETAILS

We use ViT-S/16 for both anchor and target encoders and train for 100 epochs with batch size 64 using AdamW. The initial learning rate is 0.0002 with a 15-epoch warm-up and cosine decay; weight decay follows a cosine schedule from 0.04 to 0.4. We set $\mathcal{T} = 1024$ prototypes (see Appendix A.1 for ablation), representation dimension $\mathcal{D} = 256$, loss weights $\lambda_1 = \lambda_2 = 1$, and anchor masking ratio $\alpha = 0.15$. Anchor views follow Assran et al. (2022) with one random and ten focal crops ($V_a = 11$). The target encoder is updated via EMA with momentum from 0.996 to 1.0.

## 4 EXPERIMENTS

### 4.1 TRAINING DATA OF CWA-MSN

For the development of CWA-MSN, we utilized Bray dataset (Bray et al., 2016) which encompasses five-channel cell painting images perturbed by diverse small-molecules. First, we applied the pre-processing pipeline described in Sanchez-Fernandez et al. (2023). Then, following CellCLIP (Lu et al., 2025), we selected 70% of the total data for training, including 198,609 cell images with 7,401 distinct perturbations. Note that the size of the training data (that is, 0.2 M) is much smaller than that of recent self-supervised methods (from 5M to 93M; see Fig. 1).

## 4.2 BENCHMARKS

**Gene-Gene Interaction Benchmark** RxRx3-core (Kraus et al., 2025) is a curated benchmark dataset to evaluate zero-shot performance of a cell painting image encoder, circumventing the limitations of existing benchmarks (Chandrasekaran et al., 2024; Arevalo et al., 2024a) such as small perturbation coverage and biased well positions. The dataset consists of 1,335,606 images perturbed by 736 gene knockouts and 1,674 small-molecules.

In the RxRx3-core gene–gene interaction benchmark (Celik et al., 2024), models are evaluated by computing pairwise cosine similarities between features of all gene–gene pairs (e.g., MTOR–TSC2), selecting the top and bottom 5% similarities, and comparing these predicted positive and negative interactions against curated databases, including Reactome, HuMAP, SIGNOR, StringDB, and CORUM (Giurgiu et al., 2019; Drew et al., 2017; Gillespie et al., 2022; Szklarczyk et al., 2021). Performance is measured using recall (discovered / known interactions) for each database.

In Section 5.1, we compare the proposed CWA-MSN with handcrafted features (CellProfiler (Stirling et al., 2021)), weakly supervised methods (SupCon (Khosla et al., 2020), MolPhenix (Fradkin et al., 2024), CLOOME (Sanchez-Fernandez et al., 2023), CellCLIP (Lu et al., 2025)), contrastive learning (SimCLR (Chen et al., 2020)), and self-supervised approaches (OpenPhenom, Phenom-1 (Kraus et al., 2024), and Phenom-2 (Kenyon-Dean et al., 2024)). We additionally report ViT/S-16 baselines trained without HCS data (ViT-ImageNet) and with perturbation-label supervision (ViT-WSL). Results are summarized in Table 1, along with training data size and parameter counts to assess data and parameter efficiency; FLOPs analysis is provided in Appendix A.2.

**Compound-Gene Interaction Benchmark** The RxRx3-core compound–gene interaction benchmark evaluates a model's ability to associate gene knockouts with small-molecule perturbations by computing cosine similarity between their embeddings (Celik et al., 2024). For each compound, known target genes are ranked against random genes, and performance is measured using AUC and average precision (AP). Results are reported as the mean and standard deviation of AUC and AP across compounds, together with z-scores relative to a random baseline. Ground-truth compound–gene associations are curated from PubChem, Guide to Pharmacology, WIPO, D3R, BindingDB, US Patents, and ChEMBL (Liu et al., 2007; Zdrazil et al., 2024; Harding et al., 2024).

In Section 5.2, we compare CWA-MSN against CellProfiler, CellCLIP, OpenPhenom, Phenom-1, and Phenom-2, as well as ViT/S-16 baselines trained on ImageNet-1K and the Bray dataset. We exclude CLOOME and MolPhenix due to unavailable source code and reported metrics.

## 4.3 VALIDATION OF CWA-MSN

**Batch-Effects Probing** Batch effects in image-based profiling are commonly quantified by measuring how well technical metadata (plate, batch, acquisition day) can be recovered from embeddings. In Section 5.3, we followed this standard practice by probing the recoverability of plate identity via linear classifiers and $k$NN ($k$=5) with 5-fold cross-validation. We evaluated on the full RxRx3-core dataset as well as a variant with all negative controls removed.

**Single-Well vs. Cross-Well Alignment** One of the key innovations in CWA-MSN is to utilize cross-well images as implicit data augmentation for training. In Section 5.4, we validated the effect of this cross-well sampling strategy, by changing it to a conventional single-well sampling method. We performed the comparison between single-well and cross-well based on the gene-gene interaction benchmark, using the Bray dataset for training.

**Masked Siamese Network vs. Masked Autoencoder** We checked whether a masked siamese network has indeed benefits over the popular alternative, masked autoencoder. Specifically, we adopted CropMAE (Eymaël et al., 2024), which uses pairs of cropped images as anchor and target, but optimizes masked reconstruction instead of prototype alignment.

In Section 5.5, we tested CropMAE with single-well and cross-well settings, comparing their performance with that of CWA-MSN. To examine the performance and training efficiency together, we reported not only the gene-gene interaction benchmarks but also the training time on Bray dataset.

**Masking vs. No Masking** Given that CWA-MSN is trained with weak supervision from perturbation labels, we performed an ablation study to evaluate the contribution of the asymmetric masking mechanism. In Section 5.6, we compared our masked-reconstruction scheme against a no-masking variant. The two training settings are identical except for the masking rate, which is set to zero for the no-masking model.

## 5 RESULTS

### 5.1 GENE-GENE INTERACTION BENCHMARK

Table 1: Gene–gene interaction benchmark results of different methods. *: Values from Lu et al. (2025). **: Not publicly available. N.A.: Not available.

| Training Dataset | # Images | # Perturb. | Parameters | Method | CORUM ↑ | hu.MAP ↑ | Reactome ↑ | StringDB ↑ |
|---|---|---|---|---|---|---|---|---|
| - | - | - | - | Random | .107 | .111 | .107 | .115 |
| ImageNet-1K | 1M | - | 22M | ViT-ImageNet | .342 | .420 | .144 | .305 |
| - | - | - | - | CellProfiler | .361 | .444 | .160 | .330 |
| Bray *et al.* | 0.2M | >7K | 22M | SupCon | .242 | .271 | .123 | .224 |
| Bray *et al.* | 0.2M | >7K | 22M | ViT-WSL | .249 | .290 | .148 | .242 |
| Bray *et al.* | 0.2M | >7K | 36M | MolPhenix* | .262 | .306 | .142 | .241 |
| Bray *et al.* | 0.2M | >7K | 25M | CLOOME* | .328 | .406 | .135 | .278 |
| Bray *et al.* | 0.2M | >7K | 1,477M | CellCLIP | .354 | .416 | .145 | .307 |
| Bray *et al.* | 0.2M | >7K | 22M | SimCLR | .256 | .290 | .137 | .239 |
| RxRx3+cpg0016 | >10M | >116K | 25M | OpenPhenom | .300 | .352 | **.158** | .281 |
| RPI-93M | 93M | ∼4M | 307M | Phenom-1** | .395 | .482 | .188 | .349 |
| PP-16M | 16M | N.A. | 1,860M | Phenom-2** | .486 | .553 | .197 | .415 |
| Bray *et al.* | 0.2M | >7K | 22M | **CWA-MSN (Ours)** | **.386** | **.447** | **.158** | **.327** |

As shown in Table 1, CWA-MSN outperformed all handcrafted, weakly supervised, contrastive learning methods on the benchmark gene-gene interaction, except a few large-scale private models (i.e., Phenom-1 and Phenom-2). In particular, it surpassed the SOTA weakly supervised contrastive learning method, CellCLIP, with significant performance gaps (e.g., CORUM: .354 for CellCLIP vs. .386 for CWA-MSN). Additional analysis in Section 5.3 further verifies that our performance gains indeed stem from batch-effect mitigation. Considering that the same Bray dataset was used for CellCLIP and CWA-MSN training, these results demonstrate the superior parameter efficiency of CWA-MSN with a much smaller model size (1,477M for CellCLIP vs. 22M for CWA-MSN).

Furthermore, CWA-MSN outperformed OpenPhenom, which is publicly available SOTA self-supervised method, in most of the retrieval tasks (CORUM: .300 vs. .386, hu.MAP: .352 vs. .447, and StringDB: .281 vs. .327)). The results indicate better data efficiency for CWA-MSN compared to OpenPhenom, even with the large gap between the number of training images (>10M for OpenPhenom vs. 0.2M for CWA-MSN). Additionally, its superior computational efficiency in terms of FLOPs is reported in Appendix A.2.

The benchmark results for Phenom-1 and Phenom-2 are in fact better than those for CWA-MSN. However, CWA-MSN has significantly superior data and parameter efficiency over Phenom-1 and Phenom-2 (e.g., see (b), (c) and (d) in Fig. 1). Also, it should be noted that the training data (RPI-93M and PP-16M) and source codes of Phenom-1 and Phenom-2 are not publicly available.

### 5.2 COMPOUND-GENE INTERACTION BENCHMARK

Fig. 3 shows the graphs of the compound-gene interaction benchmark for each method, reporting the AUC-ROC and AP values over the concentration. In general, the graphs of CWA-MSN and OpenPhenom are competing with each other as the top performing method. For example, CWA-MSN consistently outperforms all other methods in the AUC-ROC graph within a range of 0.25 $\mu$ Mol up to the maximum concentration, whereas OpenPhenom dominates in the other range of concentrations (see Fig. 3).

As shown in Table 2, if we closely investigate the z-scores of each method at the maximum concentration, OpenPhenom achieved the highest z-scores in both the AP and AUC-ROC metrics (3.89 and 3.16) compared to the second-best z-scores of CWA-MSN (3.55 and 2.88). Although the z-scores of CWA-MSN are slightly lower than those of OpenPhenom, these two methods possibly

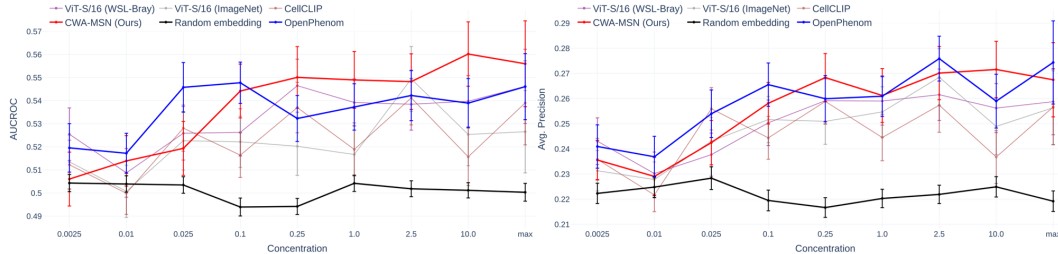

Figure 3: Compound-gene interaction benchmark graphs. AUC-ROC and AP values are reported over concentration.

Table 2: Compound-gene interaction benchmark results at the maximum concentration level. The best performance is in **bold**, and second best is in underline. *: Not publicly available. **: Evaluated using open models.

|  | AP | | | AUC-ROC | | |
|---|---|---|---|---|---|---|
| **Method** | **Mean** | **Std.** | **Z-score↑** | **Mean** | **Std.** | **Z-score↑** |
| Phenom-2* | .307 | .015 | 6.04 | - | - | - |
| Phenom-1* | .290 | .017 | 4.35 | - | - | - |
| OpenPhenom** | .274 | .017 | **3.89** | .546 | .014 | **3.16** |
| ViT-WSL | .259 | .013 | 3.37 | .546 | .016 | 2.79 |
| CellProfiler | .276 | .018 | 3.34 | - | - | - |
| CellCLIP** | .257 | .015 | 2.81 | .539 | .018 | 2.12 |
| ViT-ImageNet | .256 | .015 | 2.75 | .527 | .018 | 1.46 |
| Random | .214 | .003 | 0.00 | .500 | .004 | 0.00 |
| CWA-MSN (ours) | .267 | .015 | 3.55 | .556 | .019 | 2.88 |

have complementary strengths. For example, the std. of AP is slightly lower in CWA-MSN (i.e., better feature consistency among known relationships), whereas the mean AP is marginally higher in OpenPhenom (i.e., better capturing known relationships on average).

Most importantly, we want to highlight that this competitive performance of CWA-MSN relative to OpenPhenom was achieved despite training in a significantly (over $50\times$) smaller data size.

## 5.3 BATCH-EFFECTS PROBING

Table 3: Five-fold cross-validation results for predicting plate identity from learned embeddings (in macro-F1 scores).

| Embedding | Linear | | KNN | |
|---|---|---|---|---|
|  | **Full↓** | **No Ctrl↓** | **Full↓** | **No Ctrl↓** |
| OpenPhenom | $27.07\% \pm 0.55\%$ | $28.32\% \pm 0.37\%$ | $26.83\% \pm 0.15\%$ | $27.23\% \pm 0.46\%$ |
| CWA-MSN (Ours) | $\mathbf{13.22\% \pm 0.64\%}$ | $\mathbf{13.99\% \pm 0.85\%}$ | $\mathbf{13.32\% \pm 0.22\%}$ | $\mathbf{13.34\% \pm 0.33\%}$ |

As summarized in Table 3, across all probing strategies, CWA-MSN yields significantly lower plate-predictability (approximately half) than OpenPhenom, indicating substantially weaker entanglement with batch-specific artifacts. These results prove that the cross-well alignment strategy effectively suppresses technical variation while preserving morphological signal.

## 5.4 SINGLE-WELL VS. CROSS-WELL ALIGNMENT

As summarized in Table 4, when we tested the effect of single-well and cross-well sampling strategies combined with a masked simease network, we observed significant performance gaps between the two models. Concretely, compared to the single-well alignment (Single-Well-MSN in Table

Table 4: Gene-gene interaction benchmark results between single-well and cross-well masked simease networks. The best performance is highlighted in **bold**.

| Model | | CORUM | hu.MAP | Reactome | StringDB |
|---|---|---|---|---|---|
| | *# relationships* | *1,209* | *958* | *569* | *1,737* |
| Random | | .107 | .111 | .107 | .115 |
| Single-Well-MSN | | .281 | .330 | .130 | .261 |
| CWA-MSN (Ours) | | **.386** | **.447** | **.158** | **.327** |

4), the cross-well alignment (CWA-MSN in Table 4) largely improves recall in all gene-gene association databases. These findings show that cross-well sampling consistently outperforms the single-well counterpart in biological relationship retrieval.

## 5.5 MASKED SIAMESE NETWORK VS. MASKED AUTOENCODER

Table 5: Gene–gene interaction benchmark comparison of CWA-MSN and CropMAE. The best performance per metric is highlighted in **bold**.

| Training Time (GPU hours) | Model | | CORUM | hu.MAP | Reactome | StringDB |
|---|---|---|---|---|---|---|
| | | *# relationships* | *1,209* | *958* | *569* | *1,737* |
| - | Random | | .107 | .111 | .107 | .115 |
| 109 | CropMAE-Single | | .338 | .408 | .137 | .303 |
| 14 | CropMAE-Cross | | .348 | .443 | .135 | .309 |
| **<9** | CWA-MSN (Ours) | | **.386** | **.447** | **.158** | **.327** |

Table 5 shows that CWA-MSN consistently surpasses CropMAE (Eymaël et al., 2024) with either single-well or cross-well settings in gene-gene relationship retrieval tasks. In detail, compared to CropMAE with cross-well sampling (i.e., CropMAE-Cross), CWA-MSN achieved higher recall in all gene-gene interaction databases with the minimum training time . The results indicate that applying cross-well alignment strategy to a masked siamese network (prototype-based learning) is a more effective combination than to a masked autoencoder (reconstruction-based learning) in terms of performance and training cost. Interestingly, applying the proposed cross-well sampling strategy to CropMAE alone substantially reduced training cost while also improving benchmark performance (see CropMAE-Single vs. CropMAE-Cross in Table 5).

## 5.6 MASKING VS. NO MASKING

Table 6: Ablation study results of CWA-MSN with and without asymmetric masking strategy.

| Model | CORUM | hu.MAP | Reactome | StringDB |
|---|---|---|---|---|
| No Masking | 0.354 | 0.423 | 0.147 | 0.320 |
| CWA-MSN (Ours) | **0.386** | **0.447** | **0.158** | **0.327** |

Table 6 reports the performance of CWA-MSN compared to its variant without asymmetric masking strategy. These results indicate that asymmetric masking provides a clear benefit even in the presence of perturbation supervision. By masking only the anchor view, the model is forced to rely on perturbation-relevant morphological cues rather than low-level artifacts, improving robustness under the noisy and batch-variable conditions of cell-painting data.

## 6 CONCLUSION

In conclusion, we present CWA-MSN, a simple and effective framework for representation learning of cell painting images, which can extract phenotypic changes according to chemical and genetic

perturbations with high data and parameter efficiency. By aligning embeddings of identically perturbed cells across wells using a masked siamese architecture, CWA-MSN mitigates batch effects while preserving fine-grained morphology. This yields biologically meaningful features that improve relationship retrieval across gene–gene and compound–gene, surpassing state-of-the-art public self-supervised and contrastive baselines, even under limited data and parameter budgets.

## 7 DISCLOSURE

### MEANINGFULNESS STATEMENT

A meaningful representation of life captures the true phenotypic state of cells, reflecting how they respond to genetic or chemical perturbations while disentangling technical noise and batch effects. Our work advances this goal by using cross-well alignment and masked siamese networks to learn robust, data-efficient embeddings from cell painting images. These representations preserve biologically relevant variations, enabling more accurate modeling of cellular behavior and accelerating discovery of mechanisms and therapeutic effects.

### LLM USAGE

We used large language models (ChatGPT and Claude) to assist with code design and manuscript editing. All outputs were reviewed and validated by the authors, who take full responsibility for the accuracy and originality of this work.

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

# A  Appendix

This supplement presents extended methodological details, ablations, and quantitative analyzes in support of the main text. These experiments provide further evidence for the optimal number of prototypes(A.1) and computational efficiency(A.2).

## A.1  Prototype Number Optimization

As prototype alignment plays a key role in the training of CWA-MSN, it is important to find an optimal number of prototypes that can effectively capture biological relationships between cellular images. Therefore, we optimized the number by changing the number of prototypes (256, 512, 1024, and 2048) and measuring the performance based on the gene-gene interaction benchmark.

Table 7: Optimization results for the number of prototypes in CWA-MSN based on gene-gene interaction prediction. The best performance for each metric is highlighted in **bold**.

| Number of Prototypes | | CORUM | hu.MAP | Reactome | StringDB |
|---|---|---|---|---|---|
| | *# relationships* | *1,209* | *958* | *569* | *1,737* |
| 256 | | .372 | .433 | .132 | .321 |
| 512 | | .344 | .401 | .151 | .311 |
| 1,024 | | **.386** | **.447** | **.158** | **.327** |
| 2,048 | | .369 | .438 | .141 | .314 |

The optimization results for the number of prototypes is summarized in Table 7. When we changed the number from 256 to 2,048, the best performance was achieved at the number equal to 1,024. We potentially concluded that this is a point that balances the redundancy and diversity of prototypes.

## A.2  Computational Efficiency (FLOPs Analysis)

In the main text, we show that CWA-MSN matches or surpasses prior methods under constrained data and parameter budgets, indicating better data and parameter efficiency. To further assess computational efficiency, we compare training FLOPs.

Table 8: Computational efficiency (in GFLOPs) and gene–gene interaction retrieval performance on RxRx3-core.

| Method | GFLOPs | #Params (M) | CORUM | hu.MAP | Reactome | StringDB |
|---|---|---|---|---|---|---|
| ViT-WSL | 8.79 | 22 | 0.249 | 0.290 | 0.148 | 0.242 |
| CellCLIP | 339.17 | 1,477 | 0.354 | 0.416 | 0.145 | 0.307 |
| OpenPhenom | 104.68 | 25 | 0.300 | 0.352 | 0.158 | 0.281 |
| CWA-MSN (Ours) | **23.66** | **22** | **0.386** | **0.447** | **0.158** | **0.327** |

CWA-MSN achieves state-of-the-art biological performance while maintaining a computation footprint far below large-scale alternatives. These results show the superior computational efficiency of CWA-MSN compared to the other methods.

