# OpenReview forum: "Efficient Cell Painting Image Representation Learning via Cross-Well Aligned Masked Siamese Network"
_ICLR.cc/2026/Workshop/LMRL — ICLR 2026 Workshop LMRL Poster_

### Official Review · Reviewer_Jddf · 2026-02-22
**Boarderline accept, interesting work with very strong claims**

**Rating:** 6
**Confidence:** 4

**Review:**

# Efficient Cell Painting Image Representation Learning via Cross-Well Aligned Masked Siamese Network

## 1. Summary of the Work
The paper introduces the Cross-Well Aligned Masked Siamese Network (CWA-MSN), a representation learning framework tailored for high-content screening (HCS) cell painting images. The primary objective is to extract biologically meaningful phenotypic features while disentangling them from confounding technical batch effects. The framework achieves this by integrating two main components:
* **Cross-Well Sampling:** Selecting cells subjected to the identical perturbation from different wells, plates, or batches to act as natural, implicit data augmentations.
* **Masked Siamese Network (MSN):** Utilizing a prototype-based architecture to align the latent representations of a masked "anchor" view with an unmasked "target" view.

The authors demonstrate that CWA-MSN achieves highly competitive retrieval recall on gene-gene and compound-gene interaction benchmarks (RxRx3-core). Notably, it accomplishes this while utilizing substantially less training data (0.2M images) and fewer parameters (22M) than current state-of-the-art public models such as OpenPhenom and CellCLIP.

## 2. Overall Assessment
**Recommendation: Accept (with minor revisions for clarity and baseline fairness)**

This is a strong, practical paper that tackles a long standing bottleneck in HCS, batch-effect mitigation, with impressive resource efficiency. The experimental design is rigorous, featuring thorough ablations (e.g., masking vs. no masking, single-well vs. cross-well) and a detailed computational complexity (FLOPs) analysis. While the core architectural components are not strictly novel to the machine learning community, their specific combination and application to HCS represent a highly effective domain adaptation.

## 3. Major Strengths
* **Exceptional Resource Efficiency:** The model achieves impressive benchmark results with a fraction of the computational footprint of its peers. It outperforms the 1.48B parameter CellCLIP model using only 22M parameters, and surpasses OpenPhenom using 0.2M training images compared to OpenPhenom's >10M.
* **Effective Batch-Effect Mitigation:** The cross-well alignment objective successfully suppresses technical variation. Ablation studies clearly demonstrate that CWA-MSN cuts plate-predictability roughly in half (13.22%) compared to the OpenPhenom baseline (27.07%).

## 4. Major Weaknesses & Areas for Clarification

**A. Overstatement of Methodological Novelty**
While the empirical results are impressive, the framework relies heavily on existing paradigms. The Masked Siamese Network (MSN) architecture is pre-existing, and utilizing cross-well or replicate variations as positive pairs to force batch-agnostic representations is an established heuristic in the biological representation learning community (e.g., Metadata-guided Consistency Learning (CDCL)). The authors should temper claims of designing a strictly novel architecture and properly frame this as a highly effective, domain-specific application of MSN.

**B. Ambiguity in Baseline Evaluation (The OpenPhenom Comparison)**
There is a potential inconsistency in how the baselines were evaluated. Standard practice for utilizing OpenPhenom embeddings involves post-processing steps, eg PCA+CS or Typical Variation Normalization (TVN), to align distributions and mitigate batch effects *before* downstream tasks. In Section 5.3 (Batch-Effects Probing), the authors evaluated plate-predictability directly on the "learned embeddings". It is unclear whether the OpenPhenom embeddings were properly normalized prior to this probing. If raw OpenPhenom embeddings were compared against CWA-MSN (which explicitly optimizes for a batch-invariant latent space during training), this constitutes an unfair comparison.

**C. Incomplete Discussion on Residual Batch Effects**
While CWA-MSN reduces plate-predictability to ~13%, the authors do not discuss why this batch effect remains at this level. A 13.22% macro-F1 score for plate prediction suggests that non-trivial technical artifacts persist in the latent space. The paper would benefit from stating the baseline random-guessing metric (which depends on the number of plates) and discussing what residual factors might be driving this predictability.

**D. Lack of Details on Dataset Splitting and Scalability**
The authors state they used 70% of the Bray dataset (198,609 images) following CellCLIP's methodology. However, they do not specify how this split was performed (e.g., a random stratified split, split by plate, or split by perturbation). Given the framework's reliance on cross-well alignment, the splitting strategy could drastically alter training dynamics. Furthermore, the paper does not discuss whether CWA-MSN follows standard scaling laws or if its reliance on weak perturbation labels would cause performance to plateau if scaled to millions of images.

## 5. Questions for the Authors / Suggested Revisions

1. **OpenPhenom Post-Processing:** Could you clarify the exact preprocessing and post-processing pipeline used for the OpenPhenom baseline in the batch-effects probing? Specifically, were the OpenPhenom embeddings subjected to PCA sphering or TVN before the plate identity classifiers were trained?
2. **Dataset Splitting Details:** How exactly was the 70% split of the Bray dataset generated? Was it a random split across all images, or was it stratified by plates or perturbation classes?
3. **Residual Batch Effects:** In Table 3, CWA-MSN achieves a 13.22% macro-F1 score for plate prediction. What is the theoretical random-guessing baseline for this test set, and what are your hypotheses for why this residual technical signal persists?
4. *(Discussion / Future Work Suggestion)* **Architectural Scalability and Heuristics:** The current CWA-MSN achieves high efficiency but relies heavily on standard self-supervised heuristics (EMA encoders, stop-gradients, and tuning prototype counts) to prevent representation collapse. Given the recent introduction of frameworks like LeJEPA (Balestriero & LeCun, 2025)—which provably prevent collapse using Sketched Isotropic Gaussian Regularization (SIGReg) without requiring teacher networks—have the authors considered how their cross-well sampling strategy might couple with such heuristic-free architectures in the future? A brief discussion on how enforcing a continuous isotropic latent space might compare to the current prototype-based clustering approach for preserving subtle biological phenotypes would be a valuable addition to the conclusion.

---

### Official Review · Reviewer_QMe9 · 2026-02-24
**Review of "Efficient Cell Painting Image Representation Learning via Cross-Well Aligned Masked Siamese Network"**

**Rating:** 6
**Confidence:** 3

**Review:**

Here the authors propose a method for learning batch-effect-invariant representations of cells' morphological profiles (obtained via assays like Cell Painting). In particular, the authors propose to use masked Siamese networks with an alignment term that enforces cross-well similarity in representations of cells with the same perturbation label. The authors compare their method to previous state of the art methods (e.g. CellCLIP [1], OpenPhenom [2]) on a set of standard benchmarking tasks, and find that their method performs favorably.

Learning meaningful representation of morphological profiles is of great interest to the AI4Bio community, and I believe that this paper would generate interesting discussions at the workshop. Therefore, I lean towards acceptance. However, as-is the evaluations in the manuscript are somewhat limited, only focusing on a single dataset (RxRx3-core) and a limited set of tasks. For future work I would encourage the authors to assess their method on additional datasets (e.g. JUMP-CP [3], CellPaint-POSH data from [4], etc.) and on a broader variety of tasks.

References:

[1] Lu, Mingyu, et al. "CellCLIP--Learning Perturbation Effects in Cell Painting via Text-Guided Contrastive Learning." arXiv preprint arXiv:2506.06290 (2025).

[2] Kraus, Oren, et al. "Masked autoencoders are scalable learners of cellular morphology." arXiv preprint arXiv:2309.16064 (2023).

[3] Chandrasekaran, Srinivas Niranj, et al. "JUMP Cell Painting dataset: morphological impact of 136,000 chemical and genetic perturbations." BioRxiv (2023): 2023-03.

[4] Sivanandan, Srinivasan, et al. "A pooled cell painting CRISPR screening platform enables de novo inference of gene function by self-supervised deep learning." Nature Communications (2025).

---

### Meta-Review · Area_Chair_DSHY · 2026-02-25

**Recommendation:** Accept (Poster)
**Confidence:** 4

**Metareview:**

Accept.

---

### Decision · Program_Chairs · 2026-03-02

**Decision:**

Accept (Spotlight)

**Comment:**

Please see the meta-review.